# Identification of Cross Talk between FoxM1 and RASSF1A as a Therapeutic Target of Colon Cancer

**DOI:** 10.3390/cancers11020199

**Published:** 2019-02-08

**Authors:** Thomas G. Blanchard, Steven J. Czinn, Vivekjyoti Banerjee, Neha Sharda, Andrea C. Bafford, Fahad Mubariz, Dennis Morozov, Antonino Passaniti, Hafiz Ahmed, Aditi Banerjee

**Affiliations:** 1Department of Pediatrics, University of Maryland School of Medicine, Baltimore, MD 21201, USA; tblanchard@som.umaryland.edu (T.G.B.); SCzinn@som.umaryland.edu (S.J.C.); vivekjyoti24@gmail.com (V.B.); nehasharda4@gmail.com (N.S.); fahad.mubariz@umaryland.edu (F.M.); dennismorozov98@gmail.com (D.M.); 2Department of Surgery, University of Maryland School of Medicine, Baltimore, MD 21201, USA; ABafford@som.umaryland.edu; 3Department of Pathology, University of Maryland School of Medicine, Baltimore, MD 21201, USA; TPassaniti@som.umaryland.edu; 4The Marlene & Stewart Greenebaum Comprehensive Cancer Center, University of Maryland School of Medicine, Baltimore, MD 21201, USA; 5Department of Biochemistry & Molecular Biology and Program in Molecular Medicine, University of Maryland School of Medicine, Baltimore, MD 21201, USA; 6GlycoMantra Inc., Baltimore, MD 21227, USA; hfzahmed86@gmail.com

**Keywords:** FoxM1, RASSF1A, p-YAP, metastatic colon cancer (mCRC), cancer organoids

## Abstract

Metastatic colorectal cancer (mCRC) is characterized by the expression of cellular oncogenes, the loss of tumor suppressor gene function. Therefore, identifying integrated signaling between onco-suppressor genes may facilitate the development of effective therapy for mCRC. To investigate these pathways we utilized cell lines and patient derived organoid models for analysis of gene/protein expression, gene silencing, overexpression, and immunohistochemical analyses. An inverse relationship in expression of oncogenic FoxM1 and tumor suppressor RASSF1A was observed in various stages of CRC. This inverse correlation was also observed in mCRC cells lines (T84, Colo 205) treated with Akt inhibitor. Inhibition of FoxM1 expression in mCRC cells as well as in our ex vivo model resulted in increased RASSF1A expression. Reduced levels of RASSF1A expression were found in normal cells (RWPE-1, HBEpc, MCF10A, EC) stimulated with exogenous VEGF_165_. Downregulation of FoxM1 also coincided with increased YAP phosphorylation, indicative of tumor suppression. Conversely, downregulation of RASSF1A coincided with FoxM1 overexpression. These studies have identified for the first time an integrated signaling pathway between FoxM1 and RASSF1A in mCRC progression, which may facilitate the development of novel therapeutic options for advanced colon cancer therapy.

## 1. Introduction

More than 90% of all cancer-associated deaths are caused by metastasis. Metastatic colon cancer (mCRC) is the third most lethal disease in the United States. Approximately 50% of patients will develop metastasis during the course of colon cancer and the five-year survival is only around 55% [1,2]. In early stages of the disease, cancer lesions can be removed by surgical excision. In more advanced cases cytotoxic drugs such as 5FU, oxaliplatin, and irinotecan are used as a first line of defense. However, patients often develop resistance to these drugs within 3–12 months [3]. Therefore, understanding the cellular mechanisms that regulate metastasis is vital to the development of effective cancer therapies.

Expression of oncogenic transcription factor FoxM1 and tumor suppressor RASSF1A are among the multiple genetic and epigenetic alterations in mCRC [4,5]. The FoxM1 oncogene regulates CRC metastasis by interacting with specific cofactors to either activate or repress target genes responsible for invasion and metastasis. Yes-Associated Protein (YAP) is a known transcriptional coactivator of FoxM1 [6,7] and it is normally unphosphorylated due to inactivation of the Hippo tumor suppressor pathway in CRC [8]. Once phosphorylated, YAP translocates into the nucleus and binds to Tea Domain Family Transcription factors (TEAD1-4) [9]. YAP is also considered a corepressor for transcription regulation of FoxM1. Upstream mutations of NF2 cause YAP/TEAD to directly bind the FoxM1 promoter region [9]. Additionally, previous reports have suggested that the tumor suppressive potential of YAP1 is due to its binding to TP73 and its regulation by RASSF1A leading to the expression of pro-apoptotic genes like BBC3/PUMA and BAX [10,11]. Moreover, FoxM1 is overexpressed in metastatic cells due to loss of the tumor suppressor proteins and as a result of signaling from oncogenic factors like Ras. 

Pituitary tumor-transforming gene-1 (PTTG1) is overexpressed in different cancers including colon cancer [12]. As FoxM1, abnormal transcription and expression of PTTG1 is involved in colon cancer progression and metastasis [13] and is a FoxM1 targeted gene. FoxM1 binds to PTTG1 promoter to promote PTTG1 transcription and FoxM1-PTTG1 pathway promotes colorectal cancer metastasis [9].

Inactivation of RASSF1A, a member of Ras family, occurs in a variety of metastatic cancers including colon cancer [14,15]. RASSF1A is subject to epigenetic regulation and suppression which is largely mediated through DNA methylation [16,17]. RASSF1A promoter hypermethylation has been described in a high percentage of primary renal cell cancer (91%) [18], liver cancers (90%) [9], small cell lung cancer (89%) [14], primary nasopharyngeal cancers (70%) [16], illeal tumors (69%) [17], pancreatic cancers (63%) [19], and breast cancer (60%) [20] and dozens of others [21,22]. Interestingly, epigenetic alterations in RASSF1A via promotor methylation are increase with advanced stage colon cancer from 4.8% in stage I and 28.6% in stage II to 42.9% in stage III. However, there is a decrease in methylation frequency in stage IV (23.8%) [23,24]. These results suggest that RASSF1A suppression in mCRC could be due to other factors in addition to promoter methylation. 

We have previously demonstrated that the plant metabolite andrographolide (AGP) induces colon cancer cell death due to the activation of IRE-1, an ER stress marker [19]. We also reported that AGP enhances RASSF1A expression in CRC cells and colon cancer tumor tissue and this activity is ER stress dependent. In addition, AGP inhibits angiogenesis through the down regulation of pro-angiogenic factors such as VEGF_165_, VEGFR2, FOXM1, PTTG1, and promotes tumor suppressor gene expression, including RASSF1A [20]. Here, to investigate the relationship between FoxM1 and RASSF1A and the FoxM1 transcriptional coactivator YAP, we used a variety of approaches, including in vitro cell culture and ex vivo 3D organoid culture models. We now report (i) an inverse relationship between FoxM1 and RASSF1A expression in CRC, (ii) inhibition of FoxM1 expression in mCRC cells resulting in increased RASSF1A expression, (iii) down-regulation of FoxM1 coinciding with increased phosphorylation of YAP, and (iv) downregulation of RASSF1A and increased FoxM1 overexpression. This is the first report to identify crosstalk between FoxM1 and RASSF1A, which could be used as a novel target to advance colon cancer treatment.

## 2. Results

### 2.1. Exogenous VEGF Suppresses RASSF1A Expression in Normal Epithelial Cells

Our recent findings demonstrate a relationship between angiogenesis signaling and RASSF1A expression which is supported by exogenous VEGF suppression of RASSF1A expression in plant-derived diterpene lactone, andrographolide (AGP) treated metastatic colon cancer cells [25]. To investigate the effect of exogenous VEGF_165_ on colon cancer cell metabolism, we stimulated T84 and Colo 205 with different concentration of VEGF_165_ as indicated (6 h, 24 h, and 48 h). MTT assay revealed significantly increased cell metabolism in a time and dose dependent manner (Figure 1A,B). Moreover, to determine if RASSF1A suppression is due to angiogenic induction we stimulated normal cell lines (prostate, RWPE1; lung, HBEpc; breast, MCF 10A; endothelial, EC) with different concentrations of exogenous VEGF_165_ (12.5 and 25 ng/mL), a known inducer of angiogenic signaling. Addition of VEGF_165_ decreased the level of RASSF1A despite an increased level of VEGF receptors (VEGFR1 and VEGFR2) relative to non-stimulated cells (Figure 1C–E). Endothelial cells express high levels of both VEGFR1 and VEGFR2 receptors [26] and, therefore, we only monitored RASSF1A expression in VEGF_165_ stimulated EC cells (Figure 1F). These data suggest that an inverse relationship exists between mitogenic signaling leading to angiogenesis and reduced expression of RASSF1A.

### 2.2. RASSF1A and FoxM1 Expression in Colon Cancer

An imbalance of activation of cellular oncogenes such as FoxM1 and loss of function of tumor suppressor genes such as RASSF1A promotes colon cancer progression [25]. We next evaluated RASSF1A and FoxM1 expression in paired colon tumor (CT) and normal tissue (NT) obtained from patient surgical samples. Expression of RASSF1A was evident in normal tissue but was absent in tumor tissue (Figure 2A first lane, B). The converse was observed for FoxM1 expression (Figure 2A third lane, C). We further evaluated FoxM1 and RASSF1A expression in CRC using tissue arrays containing the specturm of colon cancer stages (stage I–IV) as well as normal colon tissue (NAT). Immunohistology showed a significant increase of FoxM1 staining with progression of colon cancer stages combined with a decrease in RASSF1A expression (Figure 3A,B *p* < 0.01 to *p* < 0.001). These results are consistent with cell line analysis. 

We previously demonstrated that AGP upregulates RASSF1A in three metastatic colon cancer cell lines, in AGP treated mice bearing human colon cancer tissue cells, and in a patient-derived 3D colon cancer organoid model (PD3D) [25]. Here we evaluated FoxM1 and its transactivator PTTG1 expression in mCRC cell lines (T84, Colo 205) in response to AGP. Metastatic colon cancer cells were treated with AGP (IC_50_ = 45 µM) for 48 h, and protein and gene expression were evaluated by immunoblot and qRT-PCR. Both mCRC cell lines demonstrated a significant decrease in FoxM1 protein levels which was corroborated with mRNA level (Figure 2D–F, *p* < 0.001). mRNA of PTTG1 is also significantly downregulated in AGP treated mCRC cells (Figure 2G, T84- *p* < 0.001, Colo 205- *p* < 0.05). Taken together the data indicate an inverse relationship between FoxM1 and RASSF1A expression in mCRC.

### 2.3. Neutralized VEGF Receptors and Akt Inhibition Upregulates RASSF1A Expression 

To further investigate the relationship between angiogenesis signaling and RASSF1A expression, mCRC cells (T84 and Colo 205) were treated with VEGFR1 and VEGFR2 neutralizing antibodies. An MTT assay revealed the neutralization level for VEGFR1 and VEGFR2 was 0.25 µg/mL for T84 at 24 h and Colo 205 at 48 h (Appendix A). This concentration was used to study the molecular mechanism of VEGFR inhibition and RASSF1A regulation. Protein analysis of neutralized VEGFR1/VEGFR2 treated colon mCRC cells demonstrated a significant increase in RASSF1A expression (Figure 4A,B, *p* < 0.001 for T84 and *p* < 0.01 for Colo 205) and decrease in FoxM1 (Figure 4A,C, *p* < 0.001) expression. Previous reports have documented that PI3K/Akt activation is often associated with colorectal cancer and promotes its development [27,28]. To understand the relationship between Akt inhibition and RASSF1A upregulation, mCRC cells (T84, Colo 205) were treated with wortmannin, a phosphoinositide 3-kinase (PI3K) inhibitor, (0–1 µM) for 24 h. As expected, inhibition of PI3K by wortmannin reduced phospho-Akt levels significantly after 24 hours treatment (Figure 4D,F, *p* < 0.001). No significant change was found in total Akt level (Figure 4D). However, wortmannin induced significantly greater expression of RASSF1A in T84 and Colo 205 at the dose range of 0.25–0.75 µM (Figure 4D,E,G). The results suggest RASSF1A regulation depends on inhibition of angiogenic signaling.

### 2.4. FoxM1 Inhibition Upregulates RASSF1A Expression

We investigated FoxM1 as the potential RASSF1A regulatory element in the angiogenesis signaling pathway. mCRC cell lines were treated with different concentration (0–8 µM) of the FoxM1 inhibitor, thiostrepton (Th) for 48 h. The qRT-PCR analysis for RASSF1A transcript levels showed significantly increased RASSF1A levels in Th-treated T84 (Figure 5A, *p* < 0.01) and Colo 205 (Figure 5B, *p* < 0.001) cells relative to untreated cells. Moreover, cell lysates were evaluated for protein expression of FoxM1, RASSF1A, p-YAP, total YAP, and PTEN. A decreased level of FoxM1 was observed in Th treated cells for both cell lines and this decrease level was dose (Figure 5C) dependent. Conversely, increased levels of RASSF1A (Figure 5D, *p* < 0.001 for T84 and *p* < 0.01–0.001 for Colo 205 cells), p-YAP (Figure 5E, *p* < 0.001), PTEN (Figure 5F, *p* < 0.001 in T84 and *p* < 0.05–0.01 in Colo 205) were observed in Th treated cells compared to untreated cells. Additionally, p-YAP increases with Th treatment, consistent with reduction in total YAP levels. This suggests that p-YAP is now available for degradation and not translocation to the nucleus. Additionally, lower levels of FoxM1 are seen with Th treatment. Since YAP is a FoxM1 cofactor (in the nucleus), lower levels of FoxM1/YAP will lead to less transcriptional repression of RASSF1A and higher levels of RASSF1A protein, as is reported in Figure 5D. The increase in _RASSF1A (and the additional increase in PTEN expression) is consistent with inhibition of growth after Th treatment. Maximum increases were found at 6 and 8 μM at 48 h in both cell lysates. We extended our observations using ex vivo human colon cancer organoids treated with thiostrepton (8 μM for 24 h and 48 h). An increased level of RASSF1A was found at 24 h and 48 h (Figure 5G, *p* < 0.01 at 24 h and *p* < 0.001 for 48 h). Taken together, these data demonstrate that RASSF1A and its signal links with oncogenic inhibition.

To further determine the nature of FoxM1 activity in the regulation of RASSF1A, FoxM1 was overexpressed in T84 and Colo 205 cells with maximum expression observed at 48 h (*p* < 0.001 for Colo 205) (Figure 6A). Overexpression of FoxM1 resulted in decreased levels of RASSF1A at 48 h (*p* < 0.001). mCRC cells were also evaluated when FoxM1 was depleted with siRNA. As shown in Figure 6B, transfection with FoxM1 siRNA resulted in significantly reduced FoxM1 protein levels compared to control siRNA treated transfected cell lysates or control cells (Figure 6B, *p* < 0.001). This was correlated with a significant increase of RASSF1A levels (Figure 6B, *p* < 0.001 for T84 and *p* < 0.01 for Colo 205). Taken together, the data confirms that RASSF1A regulation is dependent on FoxM1 activity.

### 2.5. Potential Regulation of RASSF1A by FoxM1 through Regulation of Transcription Factor Activity

RASSF1A has at least 46 binding sites for transcription factors within its promoter regions [29]. Twelve of these (REST, ZNF592, MXI1, POLR2A, E2F6, EZH2, ZNF687, NR2F1, MNT, GABPA, TARDBP, ZFX) have consensus binding sites for FoxM1. Transcription of the FoxM1-oncogene is induced by phosphorylation of Akt [25]. Moreover, RASSF1A suppression occurs due to its promoter methylation. We hypothesize that FoxM1 cooperates with transcriptional corepressors to inhibit RASSF1A expression and activity (Figure 7A, dashed line). To test whether the potential regulation of RASSF1A is indeed due to FoxM1 and the involvement of any RASSF1A-promoter-binding transcription factors which may act as corepressors, we treated T84 and Colo 205 cells with different concentrations of (0–8 µM) of Th to inhibit FoxM1 and monitored the 12 associated genes and their expression by qRT-PCR. Expression of E2F6, EZH2, NF687, MNT, GABPA, TARDBP, and REST mRNA were significantly upregulated in T84 and Colo 205 cells (Figure 7D–I,L, *p* < 0.05 to *p* < 0.001). However, a greater expression of MX-1, POLR2A and ZF592 were found in either Colo 205 (Figure 7B,C) or T84 (Figure 7K) cells. Interestingly a significantly decreased level of ZFX mRNA was found in both Th-treated cells (Figure 7J). These results are consistent with regulation of RASSF1A through the involvement of possible shared coactivator or corepressor factors, for FoxM1 and RASSF1A.

## 3. Discussion

Despite dramatic reductions in CRC incidence and morbidity over the past few decades, colorectal cancer (CRC) is the third most diagnosed cancer, and still the third leading cause of death in the United States [30]. Metastatic CRC (mCRC) accounts for most cancer-related deaths and has limited treatment options and thus results in higher mortality rates [31]. Recent efforts providing a molecular genetic analysis of human mCRC have progressed rapidly, providing a template to approach and target specific signaling pathways. As demonstrated in our previous studies, FoxM1 expression is mediated through Akt phosphorylation in the VEGF mediated angiogenesis pathway and positively associated with the progression of CRC [25]. Little is known regarding the potential use of angiogenic inhibitors to induce RASSF1A expression. Inactivation of Akt, as previously studied, facilitates the demethylation of the RASSF1A promotor and restores its expression in prostate cancer cells [32]. Our novel report has demonstrated that suppression of angiogenic signals using AGP promotes RASFF1A expression [25]. 

In accordance with our observations and to further study the molecules that influence RASSF1A expression, we used different angiogenic inhibitors—wortmannin to inhibit Akt activation, neutralization of VEGF receptors, and a FoxM1 inhibitor, thiostrepton. However, we used wortmannin in a dose dependent manner (0.25–1 µM) as recommended by the vendor, Cell Signaling Technology (Danvers, MA, USA) (#9951), to determine its effect on RASSF1A expression. Additionally, colon cancer cell IC50 (HT29) is 24 nM for wortmannin [33]. However, we agree that off-target effects are possible.

The previous observations are: (1) AGP regulates tumor suppressor gene expression (RASSF1A, PTEN, and CDKN2A), which are involved in colon cancer progression, (2) RASSF1A is upregulated by suppression of angiogenesis signaling and Akt activation. We now report (1) an inverse relationship in expression of p-Akt and RASSF1A in the mCRC cells treated with wortmannin, (2) an inverse relationship in expression of FoxM1 and RASSF1A in the mCRC cells treated with neutralized VEGF receptors. (3) An inverse relationship of FoxM1 and RASSF1A with increased mCRC staging, (4) Inhibition of FoxM1 expression in mCRC cells (T84 and Colo 205) results in an increase of RASSF1A expression, (5) downregulation of FoxM1 coincides with an increase in the phosphorylation of YAP, and (6) downregulation of RASSF1A coincides with FoxM1 overexpression.

In addition to our studies, there have been other findings investigating the relationship between RASSF1A and YAP, a key transcriptional co-factor regulating Epithelial Mesenchymal Transition (EMT) and tumor survival [34]. Conditional expression of oncogenes in mouse lung cancer models crossed with RASSF1A knockout mice showed YAP association with RASSF1A methylation, providing the optimal system to determine the metastatic potential of this alteration in a physiologically relevant setting. Moreover, YAP has been shown to be able to directly induce the transcription of CCND1 and FoxM1 via the YAP-TEAD binding site in the FoxM1 promotor region in malignant mesothelioma cells [35]. In accordance with previous studies, we observed a crosstalk between FoxM1, YAP, and RASSF1A. As previously described, when treated with thiostrepton, RASSF1A is upregulated and YAP is phosphorylated in metastatic colon cancer cells in a dose dependent manner. Going forward, using siRNA in mCRC cells, we depleted FoxM1 levels, which resulted in an increase of RASSF1A expression, suggesting a novel role of FoxM1 and its effect on RASSF1A regulation.

Since our present study demonstrates that FoxM1 is a regulator of RASSF1A, we speculated that its inhibition could alter the expression of genes who share the common binding sites for the FoxM1 transcription factor. To this end, we found ZFZ may act as a corepressor while E26, EZH2, NF687, MNT, GABPA, TARDBP, and REST act as coactivators. Further investigations are necessary to delineate the molecular mechanisms and to answer whether FoxM1 directly or indirectly affects the expression of RASSF1A. The results of our study suggest that the underlying mechanism of mCRC involves the combination of FoxM1 overexpression along with the suppression of RASSF1A. To summarize, for the first time these results demonstrate the suppression of angiogenesis signaling through any one of several steps induces RASSF1A expression in metastatic colorectal cancer cells. These results also identify crosstalk between FoxM1 and RASSF1A, which could be used as a novel target to advance colon cancer treatment. However, the mechanism by which FoxM1 regulates expression and activity directly or through the YAP co-repressor in mCRC is not clear. Therefore, further investigation will be necessary to delineate its molecular interactions with FoxM1 and YAP to regulate RASSF1A expression in our lab-based patient derived organoid model and in vivo model.

## 4. Materials and Methods

### 4.1. Ethics Statement

All de-identified colon tumor tissues were obtained from patients undergoing surgery with the approval of the University of Maryland Institutional Review Board (HP-00066889). Written consent was collected from all patients from whom discarded tissue was collected which included permission to publish results. A small piece of each tumor tissue was frozen for subsequent analysis and the remaining tissue was processed for organoid cultures.

### 4.2. Generation and Propagation of Patient-Derived Organoid Cell Cultures

Organoid cells were generated and cultured as previously described [25]. The cultures were passaged when the aggregates reached a diameter of approximately 800 µm. Organoids were treated with 8 µM of thiostrepton (cyclic peptide antibiotic, FoxM1 inhibitor, Th) for 24 h and 48 h. Treated organoids and processed human tissues as standard protocol were subjected to immunoblotting.

### 4.3. Cell Culture

T84 and Colo 205 colon cancer cell lines were cultured and treated with AGP at the indicated concentrations as previously described [27,30]. To evaluate the relationship between suppression of angiogenic signal and RASSF1A expression, the colon cancer cells were treated with different concentrations of angiogenic inhibitor or neutralizing antibody. In details, T84 and Colo 205 cells were treated with neutralized VEGFR1 and VEGFR2 at the concentration of 0.25 µg/mL for 24 h (T84) and 48 h (Colo 205) whereas thiostrepton were used with different concentrations (0–8 µM) for 48 h and wortmannin (Akt inhibitor, Cell signaling, #9951) were used (0–1 µM) for 24 h. Further, to confirm the relationship between angiogenic molecule and RASSF1A expression, different normal human epithelial cells (immortalized benign prostate; RWPE1, human bronchial epithelial cells; HBEpc, immortalized breast epithelial cells; (MCF10A) and endothelial cells (EC) were stimulated with VEGF_165_ (12.5 ng and 25 ng/mL, Gemini, West Sacramento, CA #300-827P) for 24 h. RWPE1, HBEpc, and MCF10 were kindly provided by Dr. Hafiz Ahmed, whereas endothelial cells (EC) were from Dr. Antonino Passaniti’s laboratory. RWPE1 cells were maintained in keratinocyte (Gibco, #17005-042) serum-free medium supplemented with bovine pituitary extract (13028-014) and epidermal growth factor (10450-013). HBEpc cells were cultured in bronchial/tracheal epithelial cell growth medium (Cell, #511-500). MCF10A was cultured as published protocol [36]) and EC cells were cultured in EBM-2 as indicated earlier [37]. RASSF1A, FoxM1 expression and VEGF receptors (VEGFR1 and VEGFR2) were monitored by immunoblot.

### 4.4. Viability Assay

Cell viability in the presence/absence of VEGF_165_ or neutralized VEGFR1 and VEGFR2 was assessed using the MTT assay as previously described [30].

### 4.5. Immunobloting

Immunoblotting was performed as previously described [27,38]. The primary antibodies used were against RASSF1A (eBioscience, San Diego, CA, USA, #14-6888-80), p-YAP (Cell Signaling, Danvers, MA, USA Ser127, #D9W21), YAP (G-6) (Santa Cruz Biotechnology, Dallas, TX, USA, #sc-376830), p-Akt (thr308) (Cell Signaling, #9275), total Akt (Cell Signaling, #9272), p-VEGFR1/Flt-1 (Y1213) (R and D Systems, #AF4170), p-VEGFR2 /KDR/Flk-1 (Y1214) (R and D Systems, Minneapolis, MN, USA, #AF1766), total VEGFR1 (R&D systems, #AF321), total VEGFR2 (R&D systems, #AF357), PTEN (R and D Systems, AF847), FoxM1 (R and D Systems, #AF3975), GAPDH (Sigma Aldrich, St. Louis, MO, USA, #G8795). Images were captured using a Syngene G Box digital imager (Frederick, MD, USA) and results were quantified by densitometry as previously described [25].

### 4.6. Immunohistochemistry

Unstained colon cancer tissue (stages I-IV) array with cancer and adjacent normal tissue as control were obtained from US Bioma, Inc. Two identical unstained tissues (BC05012) were subjected for immunohistochemistry as previously published protocol [31,39]. The FoxM1 (Abcam, Cambridge, MA, USA, #ab180710; dilution1/50) and RASSF1A (5 µg/mL) are used as primary antibodies for overnight followed by ImmPRESS reagents for mouse and rabbit antibodies (Vector Laboratories, Burlingame, CA, USA, #MP-7402, and #MP-7401) respectively.

### 4.7. Quantitative Real-Time Polymerase Chain Reaction (qRT-PCR)

Gene expression was evaluated as previously described [30]. Primer sequences are listed in Appendix A. Relative gene expression changes were calculated using the 2^−ΔΔCT^ method, and expression normalization was accomplished using housekeeping gene GAPDH.

### 4.8. FoxM1 Overexpression and Depletion

Plasmid for FoxM1 overexpression was purchased from Addgene (FoxM1 pCW57.1 FoxM1, #68811). For FoxM1 depletion, siRNA specific for the FoxM1 gene and and a proprietary universal negative control siRNA that does not target any known gene were purchased from Sigma Aldrich (#EHU124431 and #SIC001, respectively). The FoxM1 siRNA is a mixture of siRNA sequences prepared from an enzymatic digestion of FoxM1 cDNA. T84 and Colo 205 cells were transfected with siRNA oligonucleotides using TurboFect (Thermo Scientific, #R0533) according to the manufacturer’s recommendations. Briefly, log phase T84 and Colo 205 cells (5 × 10^4^ cells/well in 24 well plates) were transfected with 1 μg siRNA in 100 μL serum free DMEM and transfection reagent and cells were analyzed for protein expression at 24 and 48 h post transfection. FoxM1 overexpression or depletion was validated by Western blot.

### 4.9. Statistical Analyses

Statistical analysis was performed with Graph Pad Prism for Macintosh 5.0c (Graph Pad Software Inc., San Diego, CA, USA). The mean S.E. was calculated by one-way ANOVA. Significance between groups was analyzed using the post hoc Tukey’s test and Bonferroni test. *p* values were considered significant if less than 0.05 and are indicated throughout using asterisks: * *p* < 0.05, ** *p* < 0.01, *** *p* < 0.001.

## 5. Conclusions

This is the first description of a role for the oncogene, FoxM1 in regulating the tumor suppressor gene, RASSF1A. We demonstrate the suppression of angiogenesis signaling in general, and FoxM1 in particular, results in the induction or increase in RASSF1A expression. This relationship is in place in multiple cancer types including colon cancer, but in normal epithelial cells from multiple tissues as well. The result generated from this study will facilitate the identification of pathway intermediates and the development of combination drug therpies which will be useful to treat advanced colorectal cancer.

## Figures and Tables

**Figure 1 cancers-11-00199-f001:**
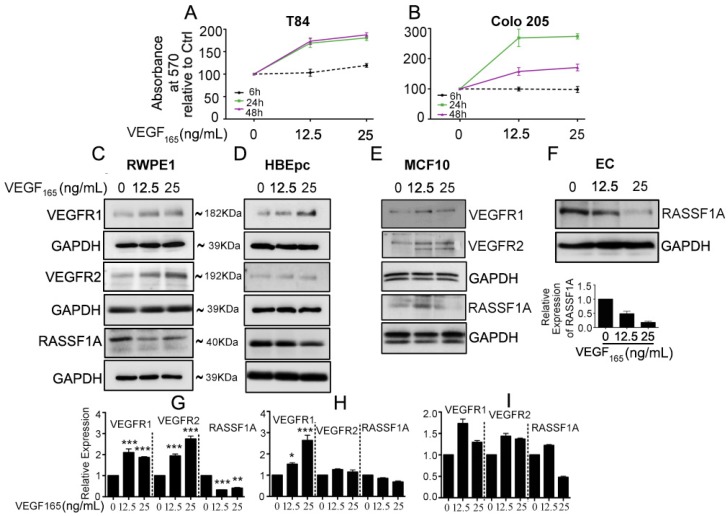
Exogenous VEGF_165_ suppresses RASSF1A expression in normal epithelial and endothelial cells. Metastatic colon cancer cells (mCRC, (**A**) T84 and (**B**) Colo 205) were stimulated with VEGF_165_ as indicated before and cell metabolism was performed by MTT assay for 6 h (black dash), 24 h (green) and 48 h (pink). Immortalized benign protate ((**C**) RWPE1), human bronchial epithelial cells ((**D**) HBEpc), immortalized human breast epithelial cells ((**E**) MCF-10A), endothelial cells (**F**, EC) were stimulated with or without VEGF_165_ (12.5 and 25 ng/mL) for 24 h. Cell lysates were monitored by Western blot for p-VEGFR1, p-VEGFR2, RASSF1A, and GAPDH as indicated. Immunoblot was quantified by scanning densitometry and normalized against GAPDH expression for RWPE1 (**G**), HBEpc (**H**), MCF 10A (**I**) and EC (lower panel of (**F**)). Results are from three independent experiments and statistical significance was determined using one way-ANOVA followed Bonferroni test. (* *p* < 0.05, ** *p* < 0.01, *** *p* < 0.001).

**Figure 2 cancers-11-00199-f002:**
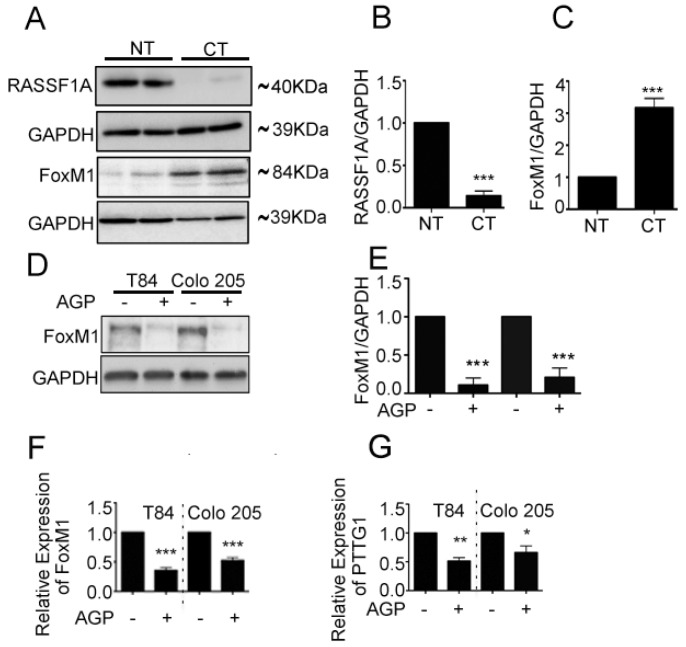
FoxM1 and RASSF1A expression is inversely co-related in colon cancer tissues. (**A**) Tissue extranction from both normal and colon cancer patient for monitoring the translational level of FoxM1 and RASSF1A by immunoblot. Quantification of RASSF1A (**B**) and FoxM1 (**C**) by scanning densitometry. GAPDH was used as a loading control. (**D**) T84 and Colo 205 cells were treated with or without AGP IC_50_ (45 µM) for 48 h. Cell lysates were analyzed by Western blot for FoxM1 and GAPDH expression. (**E**) Quantitative estimations of FoxM1 levels determined by densitometry measurements of western blots from three independent experiments after normalization with GAPDH (*p* < 0.001). T84 and Colo 205 cells were treated with or without AGP as indicated before and the transcriptional level were determined by qRT-PCR for (**F**) FoxM1 and (**G**) PTTG1. Bar graphs show quantitative results normalized to GAPDH mRNA levels. Results are from three independent experiments and statistical significance was determined using one way-ANOVA followed Bonferroni test. (* *p* < 0.05, ** *p* < 0.01, *** *p* < 0.001).

**Figure 3 cancers-11-00199-f003:**
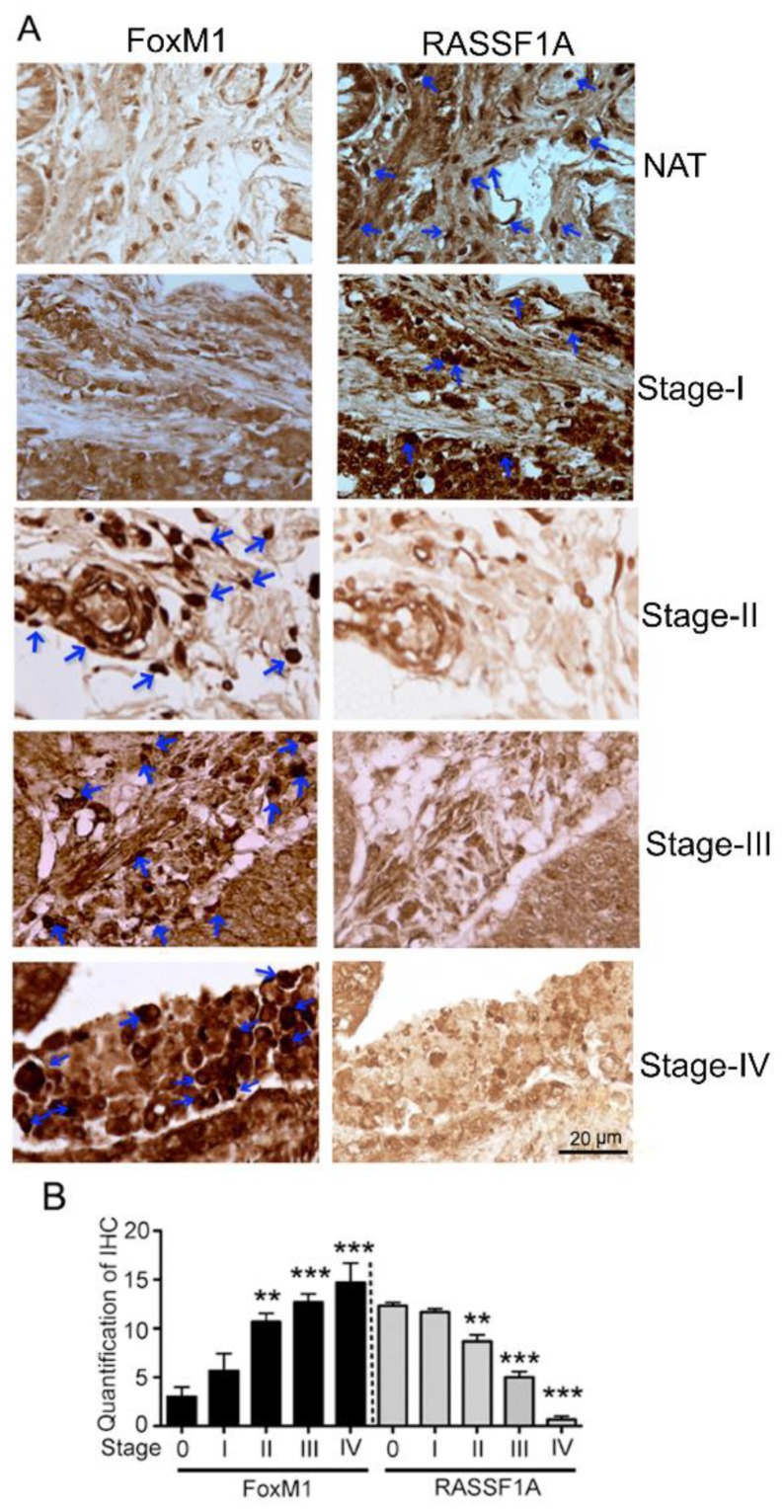
FoxM1 and RASSF1A expression in human tissue. (**A**) Immunohistochemical staining of FoxM1 and RASSF1A in different stages of primary colon carcinoma. (NAT): cancer adjacent normal colon tissue; (Stages I–IV): different stages of colon cancer tissue (400× magnification time). The histogram (**B**) represents the average percentage of FoxM1 and RASSF1A expression.

**Figure 4 cancers-11-00199-f004:**
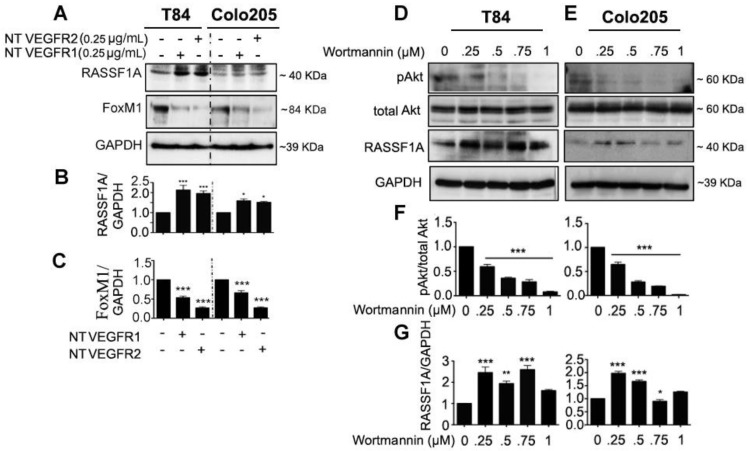
Neutralizing VEGF receptor antibodies and Akt inhibition upregulates RASSF1A and downregulates FoxM1 in mCRC. T84 and Colo 205 cells were treated with VEGF receptor 1 and VEGFR2 as indicated before (see Section 2.3). Protein expression from cell lysates was determined by immunoblotting for RASSF1A ((**A**), upper panel) and FoxM1 ((**A**), middle panel). Densitometry analysis was performed and normalyzed with GAPDH expression to demonstrate significant upregulation for RASSF1A (**B**) and downregulation of FoxM1 in the presence of neutralized (NT) VEGFR1 or VEGFR2 (**C**). (**D**,**E**) T84 and Colo 205 cells were treated with Akt inhibitor (wortmannin) with different doses (0–1 µM) for 24 h. Cell lysates were analyzed for pAkt (1st lane), total Akt (2nd lane), RASSF1A (3rd lane) expression by immunoblot analysis and quantified by densitometry (**F**,**G**). The results are from three independent experiments. (* *p* < 0.05, ** *p* < 0.01, *** *p* < 0.001).

**Figure 5 cancers-11-00199-f005:**
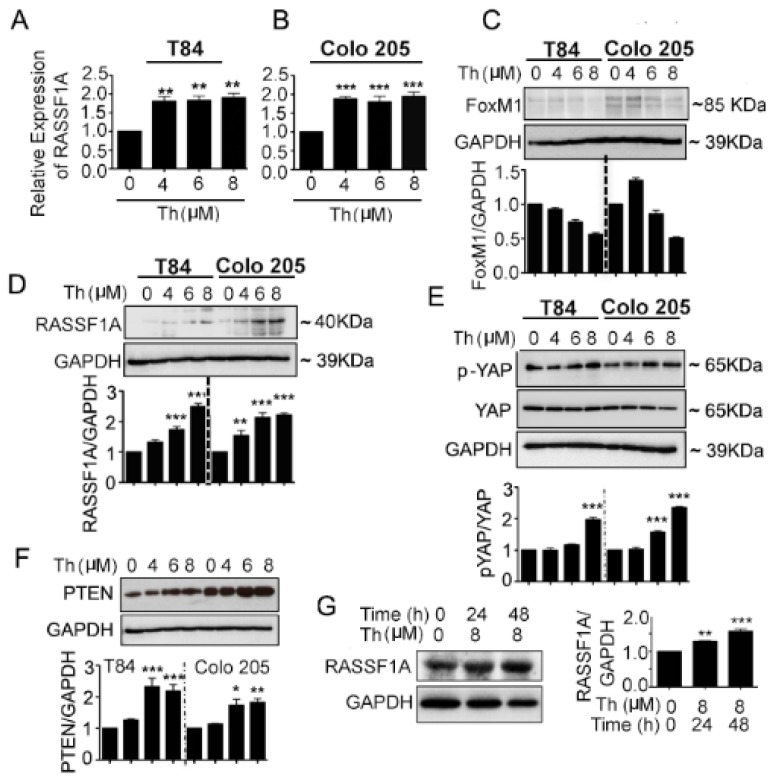
FoxM1 inhibitor induced RASSF1A and PTEN expression and YAP phosphorylation in mCRC cells. T84 and Colo 205 cells were treated with FoxM1 inhibitor, thiostrepton (Th) with different doses (0–8 µM) for 48 h. (**A**,**B**) mRNA was extracted after 24 h for detection of RASSF1A by qRT-PCR. Bar graphs show quantitative results normalized to GAPDH mRNA levels. Cell lysates were monitored by immunoblot for FoxM1(**C**), RASSF1A (**D**), p-YAP and total YAP (**E**), PTEN (**F**), and GAPDH as indicated. Immunoblots were quantified by scanning densitometry and normalized against GAPDH expression (lower panels of (**C**–**E**) for FoxM1, RASSF1A and p-YAP, respectively. Right panel of (**F**) represents the quantification of PTEN). (**G**) Patient derived organoids were treated with (8 µM) and without thiostrepton for 24 h and 48 h. Organoid cell lysates were analyzed by immunoblot for RASSF1A expression and quantified by densitometry ((**G**), lower panel). The results are from three independent experiments. (* *p* < 0.05, ** *p* < 0.01, *** *p* < 0.001).

**Figure 6 cancers-11-00199-f006:**
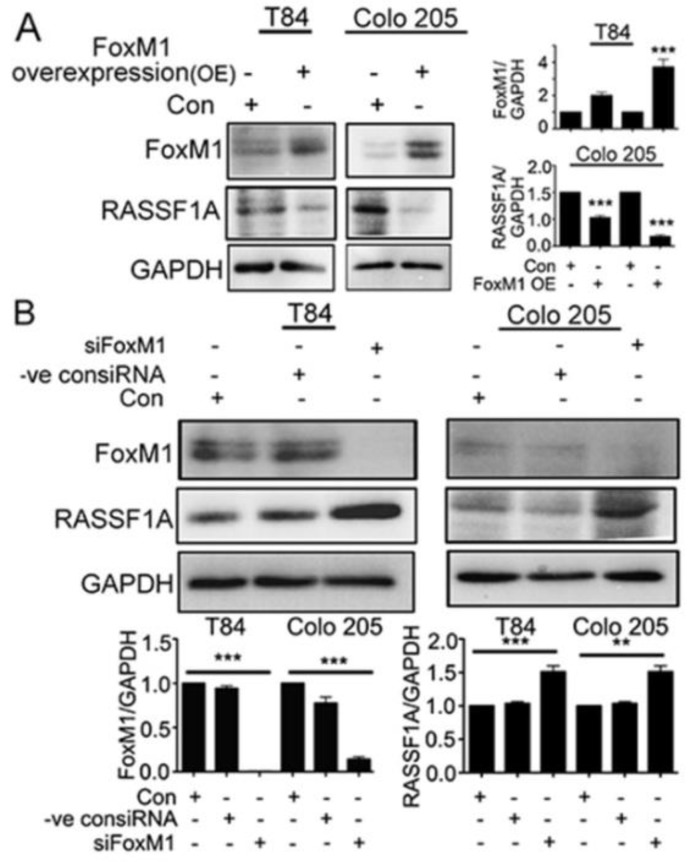
RASSF1A regulation is dependent on FoxM1. T84 and Colo 205 cells were transfected with plasmid for overexpression of FoxM1. (**A**) Cell lysates were analyzed by immunoblot and quantified by densitometry for expression of FoxM1, RASSF1A. Expression is normalized against GAPDH. The right panel of (**A**) represents the densitometric analysis of FoxM1 and RASSF1A. (**B**) T84 and Colo 205 cells were transfected with siRNA for FoxM1 or control siRNA for 48 h. Cell lysates were evaluated for FoxM1 ((**B**), 1st lane), RASSF1A ((**B**), 2nd lane) by immunoblot and quantified by densitometry. The results are from three independent experiments. (** *p* < 0.01, *** *p* < 0.001).

**Figure 7 cancers-11-00199-f007:**
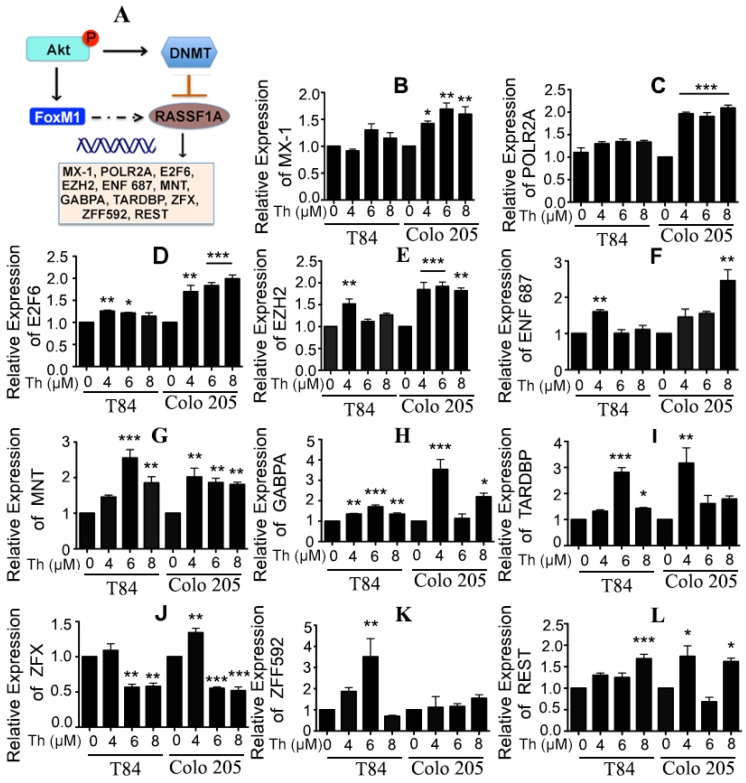
FoxM1 regulates RASSF1A through transcription factor activity. (**A**) Schematic diagram represents a crosstalk between FoxM1 and RASSF1A. (**B**) T84 and Colo 205 cells were treated with thiostrepton as indicated before and the transcriptional level of expression for RASSF1A-promoter-binding transcription factors were determined by qRT-PCR for (**B**) MX-1, (**C**) POLR2A, (**D**) E2F6, (**E**) EZH2, (**F**) ENF 687, (**G**) MNT, (**H**) GABPA, (**I**) TARDBP, (**J**) ZFX, (**K**) ZFF592, (**L**) REST. Bar graphs show quantitative results normalized to GAPDH mRNA levels. Results are from three independent experiments. Statistical significance was determined using one way-ANOVA followed Bonferroni test (* *p* < 0.05, ** *p* < 0.01, *** *p* < 0.001).

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
