# Peer review of "Identification of Cross Talk between FoxM1 and RASSF1A as a Therapeutic Target of Colon Cancer"

_cancers, 2019, doi:10.3390/cancers11020199_

Reviewer 1 Report

The ms. studied expressions of the tumor suppressor RASSF1 and the pro-oncogenic transcription factor FoxM1 in colorectal cancer. Authors show an opposite expression pattern of the two genes and suggest that FoxM1 indirectly regulates expression of RASSF1. The ms. is poorly written with missing references. Also, there are inconsistencies in the results, and mainly correlative.

Fig.1: Authors studied proliferation of colon cancer cells, but the western blots for RASSF1 are from unrelated normal cells.

Fig.2: AGP should be better explained. How does it inhibit FoxM1? How is PTTG1 related to the FoxM1/RASSF1?

Fig. 2: The IHC data is not convincing. High-mag data with size-bar should be included.

Fig. 3: Authors need to explain why Wortmanin does not increase RASSF1 at higher concentrations.

Fig.4: the purpose of including the Yap1 data in not clear. The increase is YAP-1 phosphorylation after TH treatment is not consistent with that for RASSF1.

Fig.6: It is difficult to interpret the results of TH effects on the various transcriptional protein the authors decided to assay.

Authors should try to see whether FoxM1 binds to the RASSF1 promoter by chromatin-IP.

Author Response

 Responses to Reviewers' Comments:

We are very grateful to the reviewers for their careful reading of our manuscript and their most helpful and constructive comments.  We have addressed these comments and suggestions and believe they have helped to improve the quality of this manuscript greatly.

Point 1: Fig.1: Authors studied proliferation of colon cancer cells, but the western blots for RASSF1 are from unrelated normal cells.

Our recent study demonstrated a relationship between angiogenesis signaling and RASSF1A expression by using metastatic colon cancer cells (T84 and Colo 205).  We also indicated VEGF165 treatment supersedes the effects of andrographolide (AGP) with respect to downstream angiogenesis signaling events. The addition of VEGF165 prevented the upregulation of RASSF1A expression by AGP (22). Our studies demonstrated an inverse relationship between angiogenesis signaling and RASSF1 expression in cancer cells. To further investigate the relationship between RASSF1A and angiogenic signaling we wanted to determine if this relationship was also present in non-cancer cells. In the present study, we used several normal cell lines including prostate, lung, breast, and also endothelial cells to verify that this observation was widespread. We have been unable to maintain normal colon cells in culture. These non-cancer cells typically have low angiogenic signaling and high RASSF1A expression levels. We stimulated the above-mentioned cell lines with a known angiogenic inducer, VEGF165 (12.5 and 25 ng/ml) and then determine that RASSF1A expression was reduced.  

Point 2. Fig.2: AGP should be better explained. How does it inhibit FoxM1? How is PTTG1 related to the FoxM1/RASSF1?

AGP now explained in the introduction section (Lines 79-83) and reads as “We have previously demonstrated that the plant metabolite andrographolide (AGP) induces colon cancer cell death due to the activation of IRE-1, an ER stress marker" (24). We also reported that AGP enhances RASSF1A expression in CRC cells and colon cancer tumor tissue and this activity is ER stress dependent. In addition, AGP inhibits angiogenesis through the down-regulation of pro-angiogenic factors such as VEGF165, VEGFR2, FOXM1, and PTTG1 (22).

How is PTTG1 related to the Foxm1/RASSF1?

To reflect the reviewer’s comment we added 65-68 lines in the introduction section.

It now reads “Pituitary tumor-transforming gene-1 (PTTG1) is overexpressed in different cancers including colon cancer” (9). As FoxM1, abnormal transcription and expression of PTTG1 is involved in colon cancer progression and metastasis (10) and is a FoxM1 target gene as FoxM1 binds to PTTG1 promoter to promote PTTG1 transcription and FoxM1-PTTG1 pathway promotes colorectal cancer metastasis. (9)”. In our previous study, we demonstrated that AGP downregulates FoxM1 expression. To expand our knowledge we treated colon metastatic cell lines with AGP and have shown that PTTG1 was downregulated in AGP treated cells.

Point 3: The IHC data is not convincing. High-mag data with size-bar should be included.

We have now presented our enlarged IHC data as the new Figure 3 complete with size-bar.

Point 4: Authors need to explain why Wortmanin does not increase RASSF1 at higher concentrations.

Our studies indicate that there is a dose-dependent increase in RASSF1A expression in response to wortmannin and that at higher concentrations of wortmannin, RASSF1A expression levels off.  Although RASSF1A expression did drop slightly at the highest concentration, small fluctuations at saturating doses are not uncommon.

Point 5: the purpose of including the Yap1 data in not clear. The increase is YAP-1 phosphorylation after TH treatment is not consistent with that for RASSF1.

As mentioned in our introduction, lines 56- 63 “Yes-Associated Protein (YAP) is a known transcriptional coactivator of FoxM1 and it is normally unphosphorylated due to inactivation of the Hippo tumor suppressor pathway in CRC. Once phosphorylated, YAP translocates into the nucleus and binds to Tea Domain Family Transcription factors (TEAD1-4) [6].  YAP is also considered a corepressor for transcription regulation of FoxM1. Upstream mutations of NF2 cause YAP/TEAD to directly bind the FoxM1 promoter region [6]. We are including the text  “Additionally, previous reports have suggested that the tumor suppressive potential of YAP1 is due to its binding to TP73 and its regulation by RASSF1A leading to the expression of pro-apoptotic genes like BBC3/PUMA and BAX (7,8).” In the present manuscript, we have studied an inverse relationship between oncogenic FoxM1 and the tumor suppressor RASSF1A. To the extent YAP1 may contribute to the regulation of FoxM1, YAP1 may be highly relevant to our developing model.

The increase is YAP-1 phosphorylation after TH treatment is not consistent with that for RASSF1.

In Figure 5, p-YAP increases with Th treatment, consistent with the reduction in total YAP levels. This suggests that p-YAP is now available for degradation and not translocation to the nucleus. Additionally, lower levels of FoxM1 are seen with Th treatment. Since YAP is a FoxM1 cofactor (in the nucleus), lower levels of FoxM1/YAP will lead to less transcriptional repression of RASSF1 and higher levels of RASSF1 protein, as is reported in Figure 5D. The increase in RASSF1 (and the additional increase in PTEN expression) is consistent with inhibition of growth after Th treatment. It is mentioned in the text line 207-212.  

Fig.6: It is difficult to interpret the results of TH effects on the various transcriptional protein the authors decided to assay.

In figure 6, we have tried to screen out eleven transcriptional factors of RASSF1A which have consensus binding sites for FoxM1. In a future study, we will select at least 3-4 genes which act as co-activator or co-repressor for FoxM1 to regulate RASSF1A.

Fig.7: Authors should try to see whether FoxM1 binds to the RASSF1 promoter by chromatin-IP.

We appreciate the reviewer’s thoughtful suggestion. Our manuscript is focused on establishing a relationship between angiogenic signaling and RASSF1A expression, to determine how widespread this relationship is across cells from different tissues, and among normal and cancer cells. Delineations in this pathway may aid in identifying new targets for cancer therapy. An independent study is planned to characterize the molecular bases of RASSF1A regulation.  Those studies will include the CHIP assay to investigate a possible direct interactive between RASSF1A and FoxM1.

Reviewer 2 Report

In their manuscript, Blanchard et al provide evidence for a crosstalk between FoxM1 and RASSF1A signaling. The study is well focused and the presented data clearly support the conclucions drawn.

However, Prior to publication, some issues should be addressed:

1) The title of the manuscript does not really reflect the quintessence (FoxM1 --> RASSF1A) of the manuscript and is too vague ("[...] regulates metastatic colon cancer cell Progression"..

2) PTTG1 must be introduced and put into context in the introductory part of the manuscript.

3)All "data not shown" Needs to be amended to the manuscript at least as supplementary data or not be mentioned and discussed at all.

4)Please check the manuscript (text and figures) for a consistent usage of "Akt" (and not AKt) and wortmannin (not wortmanin).

5) Figure 1: For cytotoxicity measurements, the Label "% of viability" makes sense, but in this case, where MTT is used as Surrogate Parameter for judging cell metabolism / Proliferation in the primary sense, another Label would be more precise (e.g. Absorbance at 570 relative to Ctrl, or Cell Proliferation rel. to Ctrl...).

6)Figure 2: The Arrangement of the individual Panels is very confusing! I suggest to create an additional figure with the Panels D/I focusing on the Stage-FoxM1/RASSF1A-correlation and leave the remainder in figure 2, in order to get two thematically focused figures.

7) Figure 3: Please add the concentration of the neutralizing antibody to the Label in A (Currently it is just "µg/ml"). What about the combination of NT VEGFR1 and 2? Since very high concentrations of wortmannin have been used, the authors have to comment on the specifity matter and possible off-target effects (MLCK, PI3K-related kinases) in the results/discussion (the highest concentration used is around 200x IC50!)

8) Figure 4: Again, the spatial Arrangement of the Panels is confusing (A-->B-->D-->C-->G-->E-->F in reading direction) and the Labels are overlapping (85 kDa in Panel C or G?; RASSF1A next to Panel E). Please re-arrange.

9) Figure 6: The two cell lines must be separated more clearly within the bargraphs, perhaps by adding a vertical line or by adding the x-axis Labels to each Panel (and not just to J/K/L).

Author Response

 Responses to Reviewers' Comments:

We are very grateful to the reviewers for their careful reading of our manuscript and their most helpful and constructive comments.  We have addressed these comments and suggestions and believe they have helped to improve the quality of this manuscript greatly.

Point 1: The title of the manuscript does not really reflect the quintessence (FoxM1 --> RASSF1A) of the manuscript and is too vague ("[...] regulates metastatic colon cancer cell Progression".

We thank the reviewer for this thoughtful reflection on our title. We agree that our title did not adequately relay the nature of our study. We have now changed our title to “Identification of cross talk between FoxM1 and RASSF1A as a therapeutic target of colon cancer”.

Point 2: PTTG1 must be introduced and put into context in the introductory part of the manuscript.

As described earlier (reviewer 1, Point 2, second paragraph).

Point 3: All "data not shown" Needs to be amended to the manuscript at least as supplementary data or not be mentioned and discussed at all.

The data for neutralization level for VEGFR1 and VEGFR2 has now been included in a supplementary figure (SI-1). Wortmannin data for 48 h and thiostrepton data for 24 h has not been mentioned and is deleted from the text.

Point 4: Please check the manuscript (text and figures) for consistent usage of "Akt" (and not Akt) and wortmannin (not wortmannin).

We have corrected all typographical errors regarding Akt and wortmannin.

Point 5: Figure 1: For cytotoxicity measurements, the Label "% of viability" makes sense, but in this case, where MTT is used as Surrogate Parameter for judging cell metabolism / Proliferation in the primary sense, another Label would be more precise (e.g. Absorbance at 570 relative to Ctrl, or Cell Proliferation rel. to Ctrl...).

We agree with the reviewer’s observation regarding this axis label and we have now edited the graph so that the axis label reads “Absorbance at 570 relative to Ctrl.”

Point 6: Figure 2: The Arrangement of the individual Panels is very confusing! I suggest to create an additional figure with the Panels D/I focusing on the Stage-FoxM1/RASSF1A-correlation and leave the remainder in figure 2, in order to get two thematically focused figures.

We thank the reviewer for this recommendation for new figure organization and we believe it improves the manuscript. Figure 2D and I are now represented in Figure 3. The original Figures 3-6 have changed to Figures 4-7.

Point 7: Figure 3: Please add the concentration of the neutralizing antibody to the Label in A (Currently it is just "µg/ml"). What about the combination of NT VEGFR1 and 2? Since very high concentrations of wortmannin have been used, the authors have to comment on the specifity matter and possible off-target effects (MLCK, PI3K-related kinases) in the results/discussion (the highest concentration used is around 200x IC50!)

We have now added the concentration of the neutralizing antibody to the Label in A (0.25 µg/ml). In a previous study, we tested the relationship between angiogenesis signaling and RASSF1A expression in two different colon metastatic cell lines (T84 and Colo 205) treated with andrographolide. To further investigate the relationship between angiogenesis signaling and RASSF1A expression, we treated colon cancer cells with different inhibitors such as wortmannin (Akt inhibitor), thiostrepton (FoxM1 inhibitor), and we also neutralized VEGFR1 and VEGFR2. We achieved significant changes when we inhibited each of the VEGF receptors. While there may be value in determining if simultaneous blocking provides additional increases in RASSF1A expression, we were interested in demonstrating the relative contributions of each of these receptors.

Wortmannin was used in a dose-dependent manner (0.25-1 µM) to determine its effect on RASSF1A expression. We used the concentration range recommended by the vendor, Cell Signaling Technology (#9951). Additionally, colon cancer cell IC50 (HT29) is 24nM for wortmannin. We used concentrations about 10-fold higher than IC50.  However, we agree that off-target effects are possible. Therefore, we have discussed this in the text.

Point 8: Figure 4: Again, the spatial Arrangement of the Panels is confusing (A-->B-->D-->C-->G-->E-->F in reading direction) and the Labels are overlapping (85 kDa in Panel C or G?; RASSF1A next to Panel E). Please re-arrange.

We have rearranged these panels to make for a more logical presentation.

Point 9: Figure 6: The two cell lines must be separated more clearly within the bargraphs, perhaps by adding a vertical line or by adding the x-axis Labels to each Panel (and not just to J/K/L).

We have now added labels for the two cell lines for each bar graph.

Round  2

Reviewer 1 Report

The statement that YAP1 is a co-activator or cofactor for FoxM1 is confusing, as the included reference does not provide evidence for it.

Fig.1: Panels A and B fit better as parts of Fig. 4. 

Author Response

Responses to Reviewers' Comments:

We are very grateful to the reviewer for his/her careful reading of our manuscript and his/her most helpful and constructive comments.  We have addressed first reviewer comments and suggestions.

Reviewer 1

Point 1: The statement that YAP1 is a co-activator or cofactor for FoxM1 is confusing, as the included reference does not provide evidence for it.

Response: Considering the reviewer’s comment, we have added three references and these are numbered as 6, 7 and 8.

Point 2: Fig.1: Panels A and B fit better as Parts of Fig. 4.

Response: We have decided to incorporate panel A and B in Figure 1 rather than in Figure 4 because Figure 1 demonstrates the effect of VEGF165 on colon cancer cell survival and RASSF1A expression in normal epithelial cells. As we have already demonstrated the RASSF1A expression in the presence of exogenous VEGF165 in colon cancer cells (Ref 25), the present result is an affirmation of our previously published results showing VEGF165 as a stimulator; whereas in Figure 4, we are using neutralized VEGF receptors and Akt inhibitor (wortmannin). Moreover, Figure 4 revealed the RASSF1A expression in response to neutralizing VEGF receptors and Akt inhibitor. 

Reviewer 2 Report

All issues raised by the reviewers were adequately addressed. The manuscript is now suitable for publication.

Author Response

We are very grateful to the reviewer for his/her careful reading of our manuscript. We really appreciated your time and effort.